# Reflecting Picasso in Glass

Sandrine Welte

Department of Asian and North African Studies, Ca' Foscari University, 30123 Venice, Italy; sandrinewelte@gmail.com

**Abstract:** Whereas Picasso's work in ceramics, wood and bronze is rather well known, the body of his sculptures in glass remains an object of little research. In fact, as a thorough analysis reveals, they rarely find mention in publications or catalogues on Picasso and seldom are included in exhibitions or retrospectives on the great Spanish artist. This may on the one hand be attributed to a still prevailing perception of glass as a medium for industrial, functional or everyday purposes—hence discounting the material in terms of artistic output—while on the other to controversies of authorship, related to the question of ideation versus creation. Unlike ceramics or bronze, the realisation of blown glass sculpture hinges on the involvement of the maestro vetraio as the mediator between thought and form—thus resulting in a distancing between artwork and artist conditioned by the nature of the medium. Against this background, the paper aims at a better understanding of Picasso's vision of sculpture through an examination of his creations in the vitreous medium. On these grounds, a closer look at Picasso's works in glass is meant to highlight his unique 'hand' in terms of idiom, line and form.

**Keywords:** Picasso; glass art; glass sculpture; Murano; craft; Fucina degli Angeli

## 1. Thinking Sculpture in Glass

Incredulity, wonder and adoration coalesce on Picasso's face as he lays his eyes on *Gufo*, the first of his sculptures in glass realised in collaboration with Egidio Costantini's 'Fucina degli Angeli' (Figure 1).

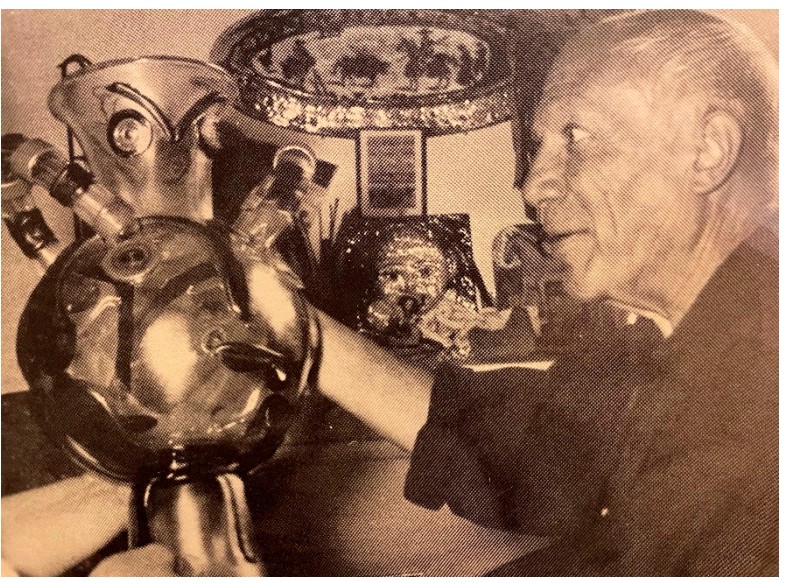

**Figure 1.** Picasso seeing *Gufo* for the first time. Credits: Attilio/Luigi Costantini. (Licht and Lader 1990, p. 37).

It is an expression of pure delight over holding the vitreous creation in his hands, the physical manifestation of an almost magical transposition of his two-dimensional drawings to three-dimensional form. Following the first essays in the 1960s, further projects take shape over the course of years that result in such iconic works as *Twenty-three glass sculptures after sketches by Picasso* (Peggy Guggenheim Collection Venice) (Figure 2), *Donna* and *Capretta* from the 'Nymphs and Fauns' series (Kunstmuseum Walter Augsburg) or *Anfora* (Fondazione Oderzo Cultura) (Figure 3).[1]

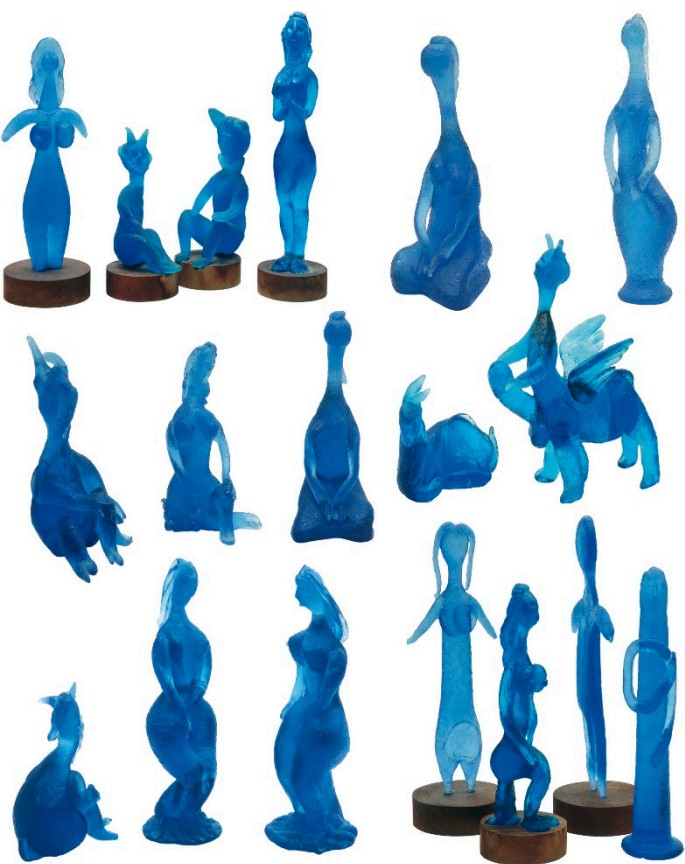

**Figure 2.** *Twenty-three glass sculptures after sketches by Picasso* (1964). Credits: Egidio Costantini (1912–2007). Poured glass. Between 10 and 30.5 cm high. Peggy Guggenheim Collection, Venice (Solomon R. Guggenheim Foundation, New York). 76.2553 PG 294.1–.23. https://www.guggenheim-venice.it/en/art/works/twenty-three-glass-sculptures-after-sketches-by-picasso/ (accessed on 21 December 2023).

Whereas Picasso's work in ceramics, wood and bronze is rather well known, the body of his sculptures in glass remains an object of little research. In fact, as a thorough analysis reveals, the vitreous creations rarely find mention in publications or catalogues on Picasso and seldom are included in exhibitions or retrospectives on the great Spanish artist. This may on the one hand be attributed to a still prevailing perception of glass as a medium for industrial, functional or everyday purposes[2]—hence discounting the material in terms of artistic output—while on the other, to controversies around authorship, related to the question of ideation versus creation. In view of the idiosyncratic properties of glass and the ways it has to be worked, these issues have informed discussions in the past with no concrete (re)solution to the present day.

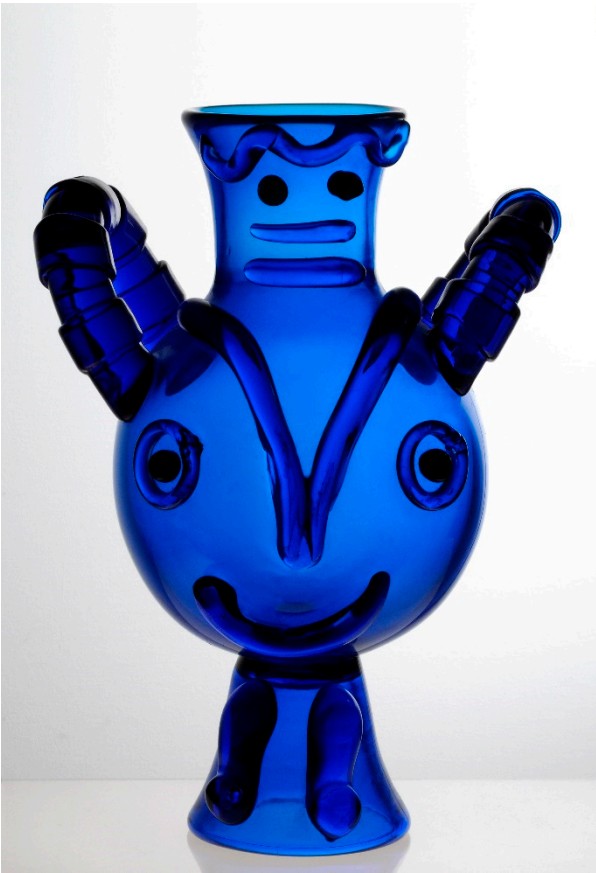

**Figure 3.** *Anfora* (1962). Credits: Fondazione Oderzo Cultura Onlus. https://www.oderzocultura.it/taglio-del-natro-per-la-collezione-zava/ (accessed on 21 December 2023).

Whilst Picasso could freely work most materials, the realisation of his blown glass sculptures required the involvement of a 'maestro vetraio', the so-called master glassmaker, thus resulting in a distancing between artwork and artist conditioned by the nature of the medium. A predicament that may however be overcome by reverting to the process of glass casting, an ancient technique which in modern times has witnessed a revival. However, as a form of potentially mechanical reproduction, it bears the risk of negatively impacting on the work's aura in the Benjaminian sense since the one-off piece is readily turned into multiples. Picasso's glass sculptures executed in both techniques accordingly lend themselves to a comparative approach that allows for a juxtaposition of artistic-manual procedures and their influence on the artist's respective iconography.

This paper seeks to address this delicate topic through Pablo Picasso's works in glass. Proceeding from a brief illustration of Egidio Costantini's 'Fucina degli Angeli' against the backdrop of the history of glass art, the vitreous medium will move to the centre of analysis in order to better understand the relationship between artist and maestro vetraio. In this context, the question of authorship will be discussed, taking into consideration the juxtaposition of artist versus artisan. Ideation as opposed to realisation shall hence become the object of a reflection on the role of either of the two figures involved in the creation of art, thereby recurring to the generally prevailing hierarchy of art over crafts.

On these grounds, the essay aims at a better understanding of Picasso's vision of sculpture through an examination of his creations in glass. The paper concludes with a closer look at Picasso's works in the vitreous medium as bearing his hand in terms of idiom, subject matter and form.

## 2. An Evolution in Glass from Craft to Art: Egidio Costantini and His 'Fucina degli Angeli'

The history of the Venetian Lagoon—and more precisely the island of Murano—has been inextricably bound to its glass-blowing tradition. From as early as the 10th century onwards, glass furnaces were operative, blending Roman experience in moulded glass with mercantile skills learnt from the interaction with the Orient (Dorigato 2002, p. 12). The earliest official document referencing the artisanship dates back to the year 982, when a certain Domenico found mentioned as a witness to a donation. In his qualification as 'fiolario', he was identified as maestro vetraio as the term 'fiole' designated bottles in blown glass, characterised by a long neck and a round body (Dorigato 2006, p. 12). By the 1200s, the craft had evolved into the city's main industrial sector, renowned for its innovative potential. Venice's position at the crossroad between East and West thereby proved conducive to establishing her as a monopoly for the manufacture and sale of quality glass. Cautiously guarded, the knowledge surrounding the production of vitreous fare constituted one of the most treasured secrets whose promulgation was punishable with expulsion from the so-called 'Serenissima', the former Republic of Venice.[3] Generations of families treasured their savoir-faire, devoted to a workmanship which for its exceptional techniques and expertise stands unique in history (Barovier 1999, p. 17). Of unparalleled perfection, the delicate creations in glass had helped the so-called Queen of the Adriatic to corroborate her pre-eminence and leading position within an economic sector that proved highly lucrative. From the earliest archaeological evidence of the trade dating back to the 7th or 8th century to its definitive transferal from Venice to the island of Murano in 1291 and its subsequent growth into a 'business sector' of global renown (Toso 2000, p. 25), the history of glass reads as a testimony to the mastery of a material whose capricious nature requires years of practice to master. Refinement and innovation lay at the heart of an artisanal tradition, unparalleled in the world.[4] Chandeliers of striking dimensions, chalices of unheard fragility and mirrors of astounding brilliance—the plethora of luxury items in the vitreous substance speaks to a gradual evolution that catered to the growing demand across Europe.

However, while glass—for centuries—was widely used for such 'design purposes', it had never been taken into consideration as a medium for artistic expression. Its properties established it within the realm of practical everyday purposes, rather than a sphere of figurative creation. Sculpture, in this regard, remained closely tied to an iconographic tradition in marble, bronze or wood. Concerned with concrete representation of the human form, sculpture as a genre in the beaux-arts witnessed its liberation from statue only in the 20th century. Gradually, it opened up to subjects other than the human figure but yet preserved its commitment to materials that had stood the test of time. This changed with the appearance of the 'Fucina degli Angeli', an ingenious venture of unprecedented kind. Spearheaded by Egidio Costantini, the initiative forged collaborations with some of the most renowned artists of the time (Barovier 1999, p. 55). It was in the early 1950s when the Venetian by choice[5] began to reach out to eminent figures such as Marc Chagall, Georges Braque or Oskar Kokoschka, asking them to hand in drawings which he executed in glass. A revolutionary idea that forever changed the course of glass, opening the doors to the realm of artistic creation.[6]

Initially working as an agent for several glass factories in Murano, Egidio Costantini gradually formulated the vision of elevating the craft of glassblowing beyond its artisanal context. Having learnt the intricacies of the trade, he was intent on proposing glass as a medium for artistic expression since he discerned its inherent potential for a new sculptural language in this material. He thus began to mediate collaborations between maestri and artists with the aim of establishing the vitreous matter for hitherto inexistent creations of great aesthetic merit and visual intrigue. Together with other Venetian artists, he formed the Centro Studio Pittori nell'Arte del Vetro in 1950 which after its dissolution in 1955 found its continuation in his 'Fucina degli Angeli', a name given by Jean Cocteau (Rizzi 1986). The so-called 'Angel's Furnace' subsequently found a temporary home in a gallery-space

in the Venetian sestiere of Castello which soon morphed into a place of artistic encounter and exchange. After a first successful foray into this unchartered territory, the gallery had to close in 1958, but with the financial aid from Peggy Guggenheim a new launch was undertaken in 1961. Following an exhibition of several glass sculptures in her Venetian palace in 1964 (Licht and Lader 1990, p. 45), a landmark was reached with Sculptures in Glass staged at the Museum of Modern Art in New York. From November 1965 until January 1966, creations by Jean Arp, Max Ernst and Pablo Picasso were on display, primarily lent by the Fucina degli Angeli but also including private collectors such as Richard L. Franck (San Francisco) as well as Nelson A. Rockefeller. It was the latter who was in the possession of 14 glass pieces by Picasso from the *Nymphs and Fauns* series.[7]

The shows marked a decisive turn for the 'Fucina degli Angeli' and over the course of time, Egidio Costantini mediated numerous fruitful collaborations with such eminent artists as Férnand Léger, Lucio Fontana, Gino Severini, and Alexander Calder (Dorigato 2002, p. 328). With the exceptional 'acqua alta' of 1966 destroying large parts of the drawings held in the gallery's archives, the 'Fucina degli Angeli' was brought to another halt until its renaissance in the 1980s.

Whereas little is known about most of the collaborations, the beginnings of Egidio Costantini's joint path with Pablo Picasso are well recorded. Together with his son[8], he travelled to Vallauris in order to meet the famous artist, who finally received him in early summer of 1954 (Licht and Lader 1990, p. 158). Picasso, eager to experiment with a hitherto unknown material, felt attracted to glass and planned on paying a visit to Venice.[9] In the meantime, the Spanish master gave a dozen drawings to Costantini for execution in the vitreous medium. In August of the same year, Costantini returned to Vallauris, bringing along the first two works, the *Gufo* and the *Flamenco* which he presented to the renowned artist (Figure 4). Upon seeing the creations for the first time, Picasso was taken by joy and enthusiasm for a material that lent a thoroughly new dimension to his sculpture. The photos taken on that occasion testify to a moment of great excitement with Picasso hugging Costantini over the realisation that a new world of artistic conception in a whole new medium had been unlocked.

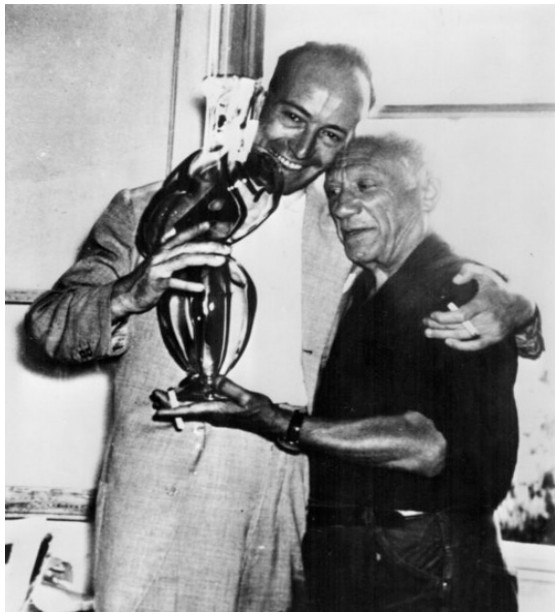

**Figure 4.** Pablo Picasso and Egidio Costantini with *Flamengo*. Credits: Attilio/Luigi Costantini (Licht and Lader 1990, p. 171).

Following this initial introduction, a prolific collaboration ensued that among others witnessed the birth of works like the *Nymphs and Fauns* series, the Centaurs as well as various vessels. An anecdote recalls how Picasso, laying his eyes on one of them (Figure 5)

for the first time, wished to sign it. Given the nature of the material and thus the inherent impossibility of doing so, he asked his wife Jacqueline for her lipstick with which he swiftly put his name on the sculpture (Rizzi 1986). Moreover, than illustrate Picasso's newly sparked passion for the medium the anecdotal account underlines one of the caveats glass comes with for artists. Its very caprice demands specific ways of handling, thus effectively conditioning the approach to it and marking it as radically different from other materials Picasso was accustomed to working in.

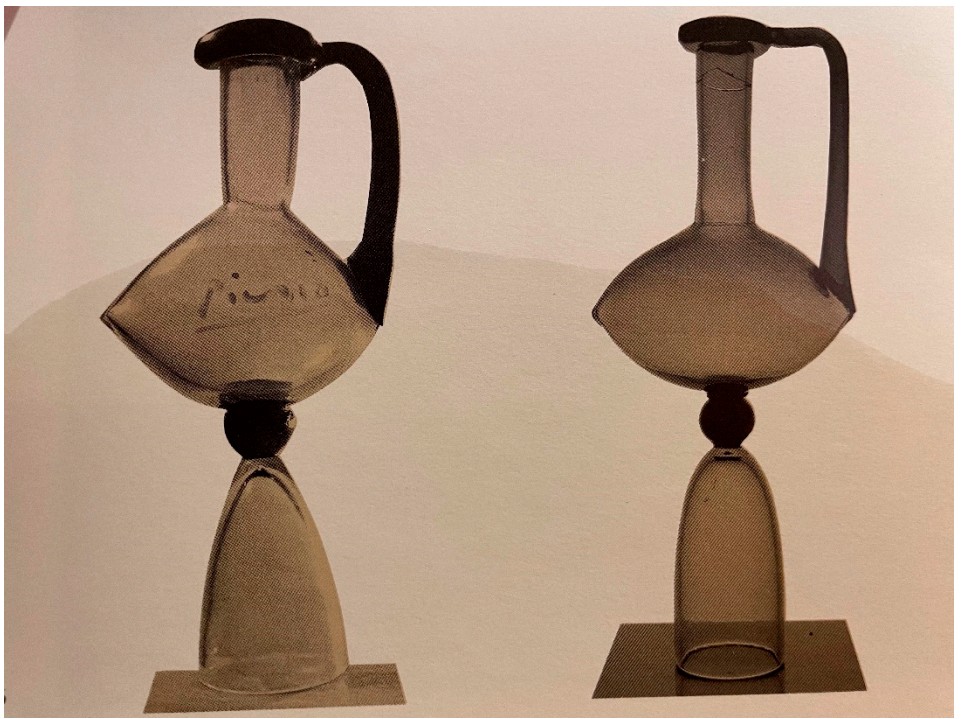

**Figure 5.** Picasso signing the *Anfora Preciosa* with lipstick. Credits: Attilio/Luigi Costantini (Licht and Lader 1990, p. 36).

### 3. Glass as a Medium for Artistic Expression

When introduced to the art world as a tool to think with and reason by, glass caused a creative revolution in the realm of sculptural representation for proposing a new poetry of pictorial reflection. Beyond traditional media—such as ceramics, marble or wood—the fiery temperament of glass invokes the genesis of its own origin, born from nature and vitrified into artefact. Dressed in countless shades, various states and copious shapes, it assumes the position of a cognitive and corporeal boundary to be transgressed towards a deeper understanding of reality, communicating the plurality of jointly inhabited worlds.

To this day, primordially associated with mundane objects of common use or pieces of decorative purpose, glass will yet have to unfold its vast potential by overcoming long-standing assumptions surrounding its emergence and evolution, towards a bolder reading that recognises its nearly endless possibilities.

In its dichotomous nature—liquid and solid, hot and cold, transparent and opaque, strong and fragile—glass hovers in a state of ambiguity and ambivalence, marked by an elusive character that defies domination. As if recalling its blazing birth in the furnace, the material refuses to be harnessed and tamed, abiding but by its very own physical properties. Working in glass, for an artist, therefore means to renounce the certainty of artistic production, conceding the process of creation to chance as well as to the hand of the maestro vetraio, while probing the heat of the furnace to intervene. The proverbial ordeal by fire accordingly comes to be at stake, challenging the artist's audacity against the quixotic, indomitable temper of glass (Barovier 1999, p. 17). From its molten state, ideas take shape in a mesmerising process that allows for no errors or corrections, demanding

instead a deliberate and decisive hand to forge its caprice into sculptural compositions (Crispoli 1984, p. 15).

In communication with its surroundings, glass ultimately comes to life through light—an immaterial counterpart that both in being reflected off and passing through the surface of the material corroborates a dual modality of perceiving vitreous creations. The gaze comes to a rest on the outer 'skin' of the work in glass while simultaneously travelling beyond, towards a realm that lies on the other side of it. At the same time, its reflective properties revert the look, a mirror drawing in the beholder as a reality of their own.

Unpredictable in its nature, glass may in this respect become a rhetorical device employed to address a panoply of issues tied to the realm of sculptural production in which artists have expressed themselves. The ambiguous nature of the material can serve to assert a creative vision by harnessing its dichotomous metaphorical qualities such as fragility and strength, transparency and opacity, malleability and brittleness for a meditation on diverse topics of global concern. In view of these broad associations with and semantics of glass, its communicative appeal has increasingly garnered attention among artists, from the very inception with the 'Fucina degli Angeli' in the 1950s to modern and contemporary declensions.

## 4. The Artist and the Maestro Vetraio

The artist's hand—a defining trait of the work—experiences a reversal through the medium of glass. Whereas other materials—such as clay or wood—allow for a physical interaction with and intervention in the matter, glass resists any form of direct manual modelling given the temperatures of more than 1000 °C in its molten state. In its amorphous state adhering to its very own laws, the vitreous matter requires years of expertise and practice in order to transform it from something widely shapeless into meaningful form. With its primary use for design and everyday objects, glass was thereby tied to a craft tradition, cementing the maestro vetraio's role and renown as an artisan. Upon its discovery as a medium for art and artistic expression however, this perception was widely challenged. Intent on harnessing the material's potential for creative means, new generations of artists turned to glass, thereby realising their dependency on the glassmith's skill—and hand—for a translation of ideas into the fragile matter. As a medium for artistic expression, glass relies on the fusion of the maestro's hand and the artist's mind, in order for a purposeful transposition to happen. Conception and execution are disjoint, unified but by the language of creative visualisation. Mutual intelligibility, an orchestration and choreography of idea and form, mind and hand, emerges as the sine qua non for the eventual genesis of a work in symbiotic operation. The act of translation from sketch to sculpture requires a shared language by maestro and artist, who jointly walk the tightrope in the creation of a three-dimensional piece whose coming into existence demands a keen awareness and profound understanding of the vitreous matter. Yet, it is the artist's vision that sparks the conception of the piece, thus marking the point of departure for an eventual creation' of the work. By this, the artist's mind becomes the innovative force, bringing form to life by guiding and harnessing the maestro's expertise. Molten sand—magnificently humming with the mysteries of a centuries-old tradition—bends under the virtuous hands of the maestro in a subtle choreography orchestrated by the artist's courageous, bold mind to morph into a sculptural paradox and modelled singularity whose vitreous soul speaks to the adventurous figurative speech of enterprising, dauntless thought. Recalling the specular properties of the medium, glass then becomes a mirror for the artist whose practice is reflected by the challenging, amorphous life of a matter that constantly calls to be tempted and pushed to its limits.

In this concerted effort, the long prevailing association of glass with design and craft is thrown into question, countering and rectifying a dominant perception while opening up to a new creative idiom in the medium which emerges at the service of artistic expression. While its inherent properties require a high level of expertise and skill acquired only over

the course of years, glass transcends its merely artisanal field of application by virtue of mediation of the artist's ideas ([Dorigato 2002](), p. 328).

The fiercely debated question of authorship with its implications derived from the imprint of the maestro's hand has prompted some artists to discard glass as a potential medium for expression. With the revival of glass casting, one of the oldest but largely forgotten techniques, the material has gained in popularity for affording a re-production of form and shape in exact copy of the initial model. Instead of relying on the individual capacities and expertise of the respective craftsman whose dexterity conditions the uncertainty of blown-sculpture, the process of casting allows for a large degree of control through the mechanical steps that replicate the method of the lost-wax technique employed for realisations in bronze.

## 5. Picasso in Glass

Changing style several times over the course of his prolific career, Picasso operated with and through a repertoire of form that stands unparalleled in the history of modern art. From the 1920s onwards, his paintings speak a new iconographic language where the human body is rendered in de- and recomposition, a constant play with perspective and multi-dimensionality.[10] What had begun as a major concern in Cubism, continued as a profound interest in his later works where the dictum of 'simultaneous vision' of the subject or object from multiple angles gave way to a pictorial idiom where these perspectival investigations merged into one single image. From the primordial concern with simultaneous vision, Picasso gradually derived a new artistic language of geometric, calculated forms whose defining trait would become the outline as a way of circumscribing and containing flat colour. The genesis of this body of works may be located in a concentration on a decomposition and collapse of the rounded, three-dimensional into a perspectival two-dimensional where the illusion of depth is conjured by the mastery of line that achieves space. Through flat expanses of colour, Picasso succeeded in conveying the impression of *volumina*[11] thereby rendering the impression of three-dimensional shapes on his canvas that seem to transcend the two-dimensional confines and flatness of his support.

It is with this iconography in mind that his sculpture has to be approached as an artistic research into the semantics of bodies as *corpo*-real presences. In view of the physical properties of glass, its marked nature as a material to unfold in space speaks to a transposition of drawn line and shape to synthesise towards a third dimension. In a comparative synopsis it becomes evident how Picasso conceives his compositions as painted sculpture in a projection of form that circumscribes and defines and thereby becomes volume.

The ductility of glass demonstrates its affinity for a volumetric–sculptural language in line with Picasso's iconography and style. The almost infinite possibilities of glass allow for experimentation with sculptural transposition apt at rendering bodies in space. An array of techniques—developed, honed and refined over the course of centuries—has contributed to an expansion of fields of use as well as languages of form. Its unlimited chromaticisms and variegated surfaces justify glass as one of the most versatile, transformable media whose changeability mark it as an ideal 'matter' for Picasso's pictorial idiom.

In comparison to works in other materials he embraced, his glass sculptures are of a more ethereal nature, characterised by the transparency and fragility of the vitreous medium. While the ordeal by fire links his creations in glass to those in ceramics, their overall visual impact is strikingly different with the latter being of an earthiness the former lack. The transparency of glass endows the sculptures with a certain lightness, affecting their very aura and presence. On these grounds, a further consideration has to be taken into account that recurs to the vast impossibility of drawing or painting on glass.[12] Decorative-figurative elements must be applied manually by the maestro who with acute precision adds sketched lines to the overall design.

The works Picasso realised in glass may be largely situated within the realm of mythology and myth with his nymphs and fauns constituting one large group of sculptures. In addition to these, bacchi and zephyrs abound, testifying to a rich heritage of folklore and

fairy tale on account of an occidental tradition of storytelling. Different from these but yet characterised by a fabled poetry are his vessels that conjure up notions of *The Book of One Thousand and One Nights (The Arabian Nights)* with Aladdin and his magic lamp being the main point of reference for the orientalising forms (Figure 6).

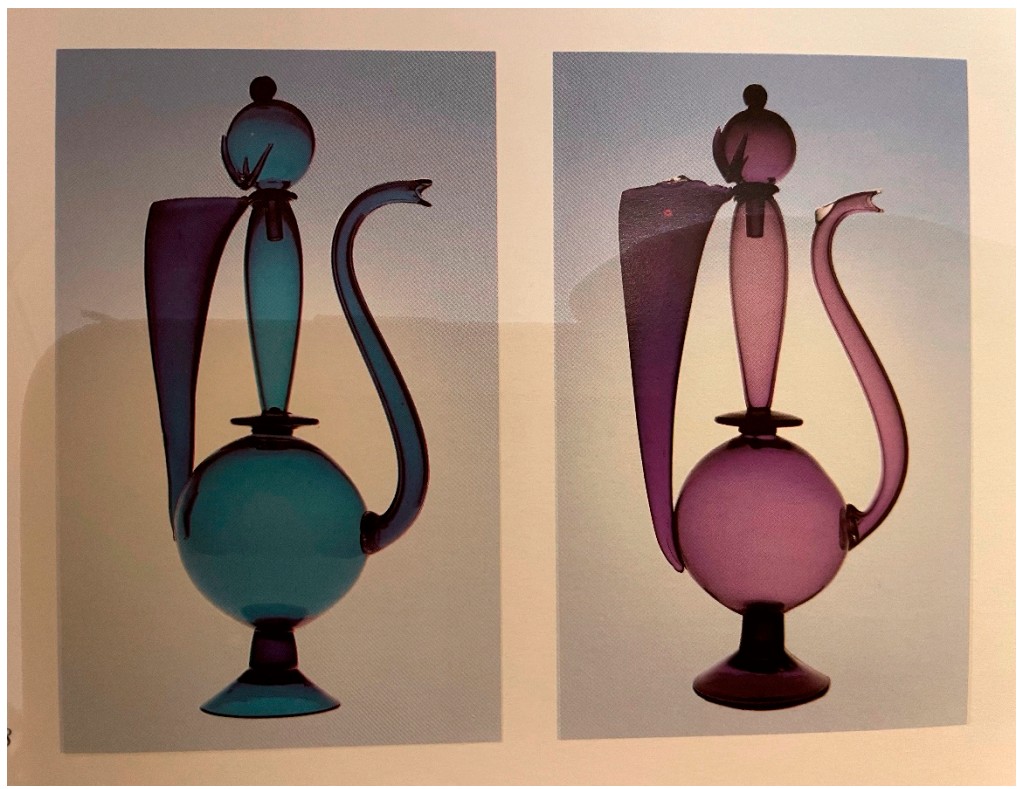

**Figure 6.** *Aquaioloa.* Credits: Attilio/Luigi Costantini (Licht and Lader 1990, p. 158).

One of the most comprehensive collection of these figurines is today held at the Peggy Guggenheim Collection in Venice where the 23 glass sculptures after his sketches form an alluring ensemble in blue poured glass. Created in 1964, their peculiar forms and shapes speak to Picasso's unique pictorial–visual language, an intriguing translation of the two-dimensional to the three-dimensional gleaned from the repertoire of characters he populated his canvases—and creative universe—with. Standing women of curvaceous silhouette alternate with their sitting animalesque male counterparts, rounded bodies that reflect his painterly idiom from the 1930s onwards. Of idiosyncratic appearance, the figures in poured glass evoke the sculptures in bronze and ceramics created by Picasso in preceding decades where they find their almost exact counterparts. The vitreous figurines held in Palazzo Venier dei Leoni thus present themselves as a continuation of his iconography albeit in a different medium. Another such example is the aforementioned *Gufo*, which as a font of inspiration found frequent iteration in Picasso's oeuvre in both painting and drawing as well as sculpture. In contrast to the blue sculptures in poured glass however, *Gufo* was executed in the traditional way of 'vetro soffiato' and hence brought to life through the maestro's breath. The irregularities of the bulbous form and undulating line are thereby the literal outpour of the medium's caprice, expressions of a matter forged into the iconic shape of Picasso's famous owl.

Whereas he was able to turn the wheel and model the clay for his works in ceramics, the translation into glass was less immediate. It thereby is interesting to see how the diametrically opposed techniques of glassblowing versus glass-casting produce optically different results. Yet, despite the differences in handling and working the medium, all sculptures invariably bear Picasso's unique 'hand'. While the maestro's expertise proves indispensable for the realisation of those three-dimensional creations in blown glass, the

result reads as uniquely and singularly attributable to the great master of the 20th century for bearing the above-mentioned iconographic traits that set his oeuvre apart. In this respect, a comparison between his two-dimensional drawn and painted works with their transposition to the haptic reality of sculpture reveals of great insight since allowing for a better understanding of Picasso's virtuous line that in essence produced volumes on paper and canvas. One such example may be found among the 23 blue sculptures in the Peggy Guggenheim Collection. In this jumble of figurines, one stands out for its striking iconographic scheme that contrasts with the rest of the 22. Seemingly astride a stud, the ithyphallic centaur recalls in its overall appearance Picasso's graphic masterpiece of *The Dream and Lie of Franco (Sueno y Mentira de Franco)*, which conceived in January 1937, depicts a similarly long-penissed political leader in the second of the 18 comic-strip-like series of images.

The immediate transposition from etching and aquatint to sculpture is most remarkable, allowing for a haptic–volumetric illustration of Picasso's graphic language. Glass as a notoriously difficult matter to handle reveals most suitable for a three-dimensional translation of Picasso's artistic 'body', a quixotic medium that in its unpredictability proves adept at rendering the line and hand of the great artist. Whereas a compatibility between both—glass and artistic line—is often not immediately given, Picasso's hand lends itself to an execution in the vitreous material, thereby preserving his gesture (and genius) in the medium while moreover breathing a new spirit into his form—albeit the maestro's own that gives life to a flat, two-dimensional design on paper in the otherwise inert substance of glass.

## 6. Conclusions

World-famed for his paintings that hold iconic status in the history of art, Picasso, whose inquisitive, curious mind moved in all directions, never limited himself to one medium but explored a vast array of materials. Correspondingly, he embraced crafts where his relentless genius found complementary modes of expression, while effectively elevating the artisanal, handiwork to the status of art. The resulting creations, irrespective of the material, thereby all form one distinctive unit that bears the imprint of their maker's singular gesture. Rather than abide by a single genre, Picasso's artistic sensibility pushed him to transgress and blur established categories, in a constant quest for new aesthetic shores. Some of his paintings thus are of voluptuous, almost sculptural quality whereas his sculptures become the support for drawings.

It was this reversal of tradition that equally drew him to glass whose unique semantics and quixotic nature exerted an immediate pull since it lent a wholly new facet to his sculptural oeuvre. The affordances of the vitreous medium for three-dimensional creation thereby presents an extensive field of research for a better understanding of Picasso's distinct idiom which yet has to come to a broader attention.

**Funding:** This research received no external funding.

**Data Availability Statement:** Data are not contained within the article. Data sharing is not applicable.

**Conflicts of Interest:** The author declares no conflicts of interest.

## Notes

[1] A Blue Centaur is furthermore held in the Carnegie Museum of Art, a similar version of which used to be in the collection of SFMoMA. https://collection.carnegieart.org/objects/4eeb578f-e4dc-4ee6-9be4-e40f84057221 (Last accessed: 24 November 2023).

[2] Another common thread of association links glass with the rather kitschy trinkets and tasteless souvenirs produced in great quantities in Venice Peggy Guggenheim herself lamented already (SEE MoMA press release for show 1965).

[3] Yet some master glassmakers could not withstand the lure of foreign empires that promising wealth and renown attracted some artisans.

[4] Whereas other centres of glass production developed independently in Bohemia or Finland, none of them lived up to the excellence pursued and achieved in Venice.

5    Born on 22 April 1912 in Brindisi, Puglia, Egidio Costantini moved to Venice at the age of six following his father's premature death and the subsequent decision to join his mother's family in the Lagoon City.

6    Today, this tradition is continued by Berengo Studio in Murano. For more than 30 years, Adriano Berengo—the spirit behind the enterprise—has been pushing the boundary of glass art, venturing in unchartered territories and advocating the use of the material for ground-breaking creations in the medium that speak to a yet unrealised potential of the vitreous matter in a contemporary context. Through his unwavering dedication to the century-old craft paired with a vision for its future he has introduced artists such as Tony Cragg, Thomas Schütte, Laure Prouvost and many more to the medium.

7    It is the same series that also captured Peggy Guggenheim's attention who added 23 of the creations to her own collection. https://www.moma.org/documents/moma_master-checklist_387318.pdf (Last accessed: 24 November 2023).

8    As Egidio Costantini's son was a professional photographer, the encounter was captured in a series of emblematic snapshots.

9    This visit most likely never took place. Whereas Picasso showed himself determined to come to the furnace in Murano, no written evidence testifying to this visit exists. A thorough analysis of Peggy Guggenheim's guest books brought forth an entry reading "P. Picasso". While this might lead to the conclusions that he did ultimately travel to Venice, it was his daughter Paloma instead, who was received by the American heiress in her palace.

10    The German term "Vielansichtigkeit", i.e., "multi-angle perspective" is most apt at rendering Picasso's modus operandi. The lack of an English equivalent thereby speaks to the predicament of ekphrasis.

11    Compare La Baignade (On the Beach) (1937) held at the Peggy Guggenheim Collection.

12    In this respect, a differentiation with stained-glass windows must be made which reposes on yet another technique and tradition.

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
