# Peer review of "Reflecting Picasso in Glass"

_arts, 2024_

Round 1

Reviewer 1 Report

Comments and Suggestions for Authors

Clear and well-researched paper, well-written.

The paper draws the reader in by a description and image of Picasso seeing one of his drawings rendered in glass, followed by a succinct introduction of its subject – Pablo Picasso’s works in glass and their public recognition and exhibition, and, by extension, the scope of glass as a medium for artistic expression, specifically for artists that are not themselves trained in glassmaking, which also raises the question of authorship. The context is established through a short history of glassmaking in Venice, the virtues of glass as an artistic medium are extolled with eloquence, and Picasso’s unique style is described. The author asserts that Picasso’s style seems to lend itself to being rendered in sculpted glass, and that Picasso’s hand is identifiable in any medium, and suggests that more knowledge is to be gained from research into Picasso’s works in glass.

Recommendation:

There could be one or two current references to well-known artists working in glass as well as other materials, such as Ai Weiwei, Olafur Eliasson, Roni Horn etc., to support the recommendation to include Picasso’s glass sculptures in publications and exhibitions and as objects of research.

Question about accuracy of museum information:

Line 36 and 337ff- (about the Twenty-three glass sculptures after sketches by Picasso in the Peggy Guggenheim Collection Venice)

 “poured glass” - this looks like hot-formed glass (or hot-worked or hot-sculpted), meaning glass from a furnace that is manipulated while it is hot. The term “Poured glass” normally refers to cast or pressed glass.

I can’t be entirely sure from the low-resolution image, but these forms would be much harder to achieve through casting than through hot-working from the furnace. I have found similar works online, where the technique is listed as “freely formed and fused together”

https://www.mutualart.com/Artwork/NINFA/EDCD6AB3E765FC043485531BE17787C8

or “melted together and freely shaped”

https://www.mutualart.com/Artwork/--SATYR-ASSIS--/D88844F790177FCA

Both of these should be hot-worked, hot-formed or hot-sculpted in English.

Also, if Picasso had modelled the figures in clay or wax, it would have made sense to make moulds and cast them, but as they are made from drawings, it is much more likely that they have been hot-formed. I assume that the correct terminology was lost in translation. 

Small niggles:

Line 38 – broken link

Line 52 – Maestro vetraio meaning? As the language of this paper is English, why not use the English term, “master glassmaker”, or provide a translation?

Line 413 – “death” is misspelled

Author Response

Hello,

Thank you very much for your helpful remarks.

I took care of the “niggles“ and fixed the link. On the corresponding webpage you will find that the blue sculptures are described as “poured“ glass. I agree that it will probably have involved some freestyle shaping; however, since the wording was gleaned from the official homepage of the Peggy Guggenheim Collection where the figurines are housed I prefer to go by this term.

As for citing contemporary artists, since the focus is on Picasso I preferred not to mention any other names in passing in the main text and instead updated the footnote on Adriano Berengo who for 35 years has been working with such eminent artists as Tony Cragg, Tracy Emin, Magdalena Campos-Pons, Ai Weiwei and many others. By citing these collaborations, a focus on the continuation of the Fucina degli Angeli in the Venetian Lagoon was intended.

Reviewer 2 Report

Comments and Suggestions for Authors

The article does not mention keywords

The article is of great importance regarding the artwork of a prodigious artist, Picasso whose glass work is not very well known. However, the methodologic structure should be better organised.

It should be a small introduction to the artist and his work, before introducing the glass sculptures. The introduction of Picasso’s work is at the end of the article, point 5. For a better organization of the article, it is recommended to rework the presentation structure.

Line 30 – ‘ flowing the first essays on the 1960s’ – what are the first essays? The author should mention the first essays

Line 120 – please explain what is “Serenissima”

Line 423 – do you mean premature death? Please speak more about Egidio Costantini . how did he become an ‘agent for several glass factories’ line 148

The legend of the figure 4 is on the top of the figure and not on the bottom

Line 218 – the author says ´therefore means to renounce the certainty of artistic production,’

This statement is not at all correct. Today, many artists make their glass pieces, and I could mention countless examples here. However, I understand what the author means, is referring to the creation of blown glass pieces where the work is done by someone other than the artist.

This sentence should be rephrased for a better understanding.

Line 243 - ´glass resists any form of manual modelling´. Do you think this is correct? The glassblowers mould the glass with the hand. Of course, not directly he uses a yet newspaper on the top of his hand but is a free-style modelling. I would recommend rewording the sentence.

Line 329 ´his vessels that conjure up notions of The Book of One 329 Thousand and One Nights (The Arabian Nights) ´ are you speaking about the work ‘Aquaioloa’, the year of production according to the legend is 1990. This is confusing…. Please explain better

Line 346 – ‘Another such example is the aforementioned Gufo,’, is it possible to provide a photo?

Author Response

Hello,

Thanks for the helpful annotations which I tried to integrate as best as possible.

Keywords were added.

The first “essays“ were the first “attempts“ such as the one cited in the article.

“Serenissima“ is the sobriquet of Venice, denoting the city as the “Most Serene Repulic“.

“Deatch“ indeed was a typo I rectified.

Lin 218 was meant as a reference to the glassblowing tradition, a skill that takes years to master and hence requires the artist to give up some of their “agency“. This is further discussed in the subsequent chapter which is why I prefer to leave it as it is.

The “manual modelling“ was further adapted to be more precise about the “direct“ intervention.

1990 is the publication date of the book where I found the image.

A photo of “Gufo“ can be found further up.